# Sleep Stage Detection from Actigraphy and Heart Rate Using an Attention-Based Model

Amyn Sorkhilalehloo
*Dept. of Electrical and Computer Engineering*
*Queen's University*
Kingston, Canada
amyn.s@queensu.ca

Nasim Montazeri Ghahjaverestan
*Dept. of Electrical and Computer Engineering*
*Queen's University*
Kingston, Canada
nasim.montazeri@queensu.ca

*Abstract*—**Sleep plays a crucial role in human well-being, while insufficient sleep affects cognitive function, decision-making, and overall health. Sleep assessment via polysomnography (PSG) is time-consuming, resource-intensive, and limited to in-laboratory sleep testing. To address the challenges of PSG, wearable sleep screening devices have been widely used, especially to detect wakefulness and sleep stages. This study proposes deep models for the detection of wakefulness versus different stages of sleep using heart rate and wrist actigraphy extracted from the multi-ethnic study of atherosclerosis (MESA) sleep dataset. First, two sets of features were extracted from heart rate and actigraphy, which were separately fed into two separate branches of convolution neural network (CNN), then merged and fed to a deep classifier. The model detected wakefulness versus sleep and different sleep stages with the accuracies of 88.19% and 79.6% respectively. This work showed that combining heart rate, actigraphy signals, and demographic data in a deep framework could improve sleep stage-staging performance. This study offers a subject-specific approach for sleep assessment based on convenient wearables.**

*Index Terms*—**Sleep, Sleep Stage, Actigraphy, Heart Rate, Deep Learning.**

## I. INTRODUCTION

SLEEP is essential for maintaining healthy cognitive functions, emotional regulation, and memory consolidation [1]. Poor sleep quality has been associated with a wide range of health problems, including cardiovascular diseases, metabolic disorders [2], mental health problems [3], and impaired immune function [4]. The gold standard for assessing sleep is polysomnography (PSG). A PSG recording is segmented into 30-second epochs, each manually annotated as wakefulness or different sleep stages including non-rapid eye movement (non-REM) and rapid eye movement (REM). However, PSG is limited by its costs, long waitlist, attachment of dozens of sensors to the head and body, in-laboratory sleep tests, and inability to monitor sleep in natural environments [5]. These limitations have led to interest in more scalable, wearable-based sleep monitoring technologies.

One of the technologies widely used for sleep monitoring is wrist actigraphy [6]. In actigraphy, wrist motion is collected using accelerometers, and thus motionless intervals are inferred as sleep, whereas intervals with motion are labeled as wakefulness. Despite extensive applications in wellness

This work was funded by Queen's University.

and clinical purposes, its dependence on wrist motion alone can lead to misclassifications during motionless wakefulness. Therefore, an additional modality, such as heart rate, can provide complementary information and enhance the accuracy of sleep staging [7].

Heart rate is higher during wakefulness compared to sleep. In REM sleep, the heart rate can be increased by the activity of the autonomic nervous system that occurs due to sympathetic bursts [8]. To measure heart rate, photoplethysmography (PPG) technology is used, which can be embedded along with actigraphy accelerometers in a wrist band. PPG uses an LED light and a photodetector in contact with the skin to track blood volume pulses, from which beat-to-beat intervals are derived [9].

Aside from detecting wakefulness versus sleep, which quantifies sleep duration, accurate breakdown of sleep into REM and non-REM stages is necessary to quantify sleep depth, which is clinically important for assessing general health and detecting specific disorders. For example, REM sleep plays a crucial role in cognitive and emotional brain function, supporting memory consolidation and emotional regulation by processing affective experiences during sleep [10], [11]. Moreover, REM duration was found to be reduced in some types of dementia [12]. Thus, detecting the depth of sleep can provide a means for detecting sleep and neurological disorders and improving sleep quality.

Early studies on wearable-based sleep staging relied on basic classifiers. Xiao et al. extracted Heart Rate Variability (HRV) features and used a random forest to classify wakefulness, non-REM, and REM sleep [13]. A study by Yuda et al. used HRV metrics and body movement as features to be fed to a multivariate logistic regression to classify sleep stages [14]. These shallow models typically achieve moderate accuracies (74.5–75.8%). Recent studies used deep learning approaches with stronger mining ability to improve sleep staging accuracy. Walch et al. applied several classifiers, including logistic regression, random forest, and multi-layer perceptron (MLP), which were fed with motion, heart rate, and clock proxy features. Their MLP model, trained on Apple Watch data, outperformed other classifiers [15]. Zhai et al. trained multiple CNN and LSTM models on the Multi-Ethnic Study of Atherosclerosis (MESA) dataset using different input

window sizes and then fused their outputs using ensemble techniques (mean-over-classifiers and max-selection), achieving their best performance over each individual baseline [16]. Song et al. used MESA, applying CNNs to extract features from raw actigraphy and heart-rate signals, then fed those into a Sequence-to-Sequence LSTM with attention to predict each sleep stage, achieving an enhanced accuracy of 79.11% [17]. Pini et al. introduced Neurobit-HRV, a deep architecture trained with RR-interval sequences extracted from ECG recordings and demonstrated that the model could perfectly generalize over age, sex, and sleep apnea groups, suggesting that ECG-derived features may serve as a reliable and lower-cost alternative to PSG [18]. However, earlier deep learning approaches were mostly developed on relatively small datasets [15]. In previous studies with larger sample sizes, such as those by Zhai et al. [16] and Song et al. [17], a single CNN branch was used to extract important features from actigraphy and heart-rate signals, which may limit its ability to learn modality-specific information.

In this work, we designed an attention-based model with specific branches for actigraphy and heart rate for two main objectives: 1) improving sleep stage detection accuracy and 2) designing a model to receive subject-specific demographics and assessing its contribution to improve the model's performance. With this study, we contributed to:

1) Proposing a novel feature extraction approach using two parallel CNN branches for heart rate and actigraphy, combining them through a linear layer to improve the representation of the extracted features.
2) Designing an encoder comprising two-layer LSTM with attention to capture temporal context and generate the sleep-stage.
3) Integrating demographic characteristics into the proposed model to evaluate their influence on the sleep-staging accuracy.

## II. METHOD

### A. Dataset

In this study, we used the MESA dataset [19], [20] to train and evaluate our model. MESA is a multicentre longitudinal study that includes a single night PSG of 2,237 participants in a wide age range (45–84 years) and diverse ethnic backgrounds (Black, White, Hispanic, and Chinese American). All participants provided their written informed consent. Participants wore an actigraphy device continuously for one week and then underwent in-laboratory single-night PSG while wearing the device, allowing simultaneous actigraphy and PSG recordings. The sleep labels were manually scored by trained technicians, ensuring high reliability and quality for research and analysis. The high-quality sleep labels and the diverse population in the MESA dataset provide an ideal foundation for training and testing our model. Table I summarizes the demographics of the included subjects.

TABLE I
DEMOGRAPHICS OF THE EXTRACTED SLEEP DATA

| Demographics | |
|---|---|
| N (Female%) | 568 (56.1%) |
| Age (years) | $69 \pm 9$ |
| Body mass index (BMI, kg/m$^2$) | $28.4 \pm 5.3$ |
| Apnea/hypopnea index (AHI, events/hr) | $18.5 \pm 17.8$ |
| Total sleep time (TST, hours) | $6.2 \pm 1.2$ |
| Sleep efficiency (SE, %) | $67.3 \pm 11.4$ |

Values are in Mean $\pm$ Standard Deviation, except for the sample size.

### B. Preprocessing

From the MESA database, 1012 subjects' data with high-quality PSG (rating $\geq 5$) and at least 6.6 hours of recordings were extracted for this study. In MESA, R peaks in ECG signals were detected using the Compumedics Somte software (v2.10). From the sequence of RR intervals, the heart rate (HR) signal was obtained. Any RR interval shorter than 0.33 s was replaced by the midpoint of that interval and the previous one. Any RR interval longer than 1.33 s was divided into $N$ equal sub-intervals ($N = T/T_{\mathrm{mean}}$), where $T$ is the current interval duration and $T_{\mathrm{mean}}$ is the mean RR interval in that epoch. From the available RR intervals, heart rate signals along with reference sleep stage scores were extracted and their alignment with actigraphy was verified and fine-tuned using cross-correlation [17]. The aligned data were then segmented into 30 s epochs, for each of which the mean and standard deviation of the heart rate were calculated and any epoch with a mean greater than two standard deviations was excluded as outliers [17].

### C. Feature Extraction

From actigraphy and heart rate signals, features were extracted over non-overlapping epochs. Actigraphy-based features included Euclidean norm minus one (ENMO) and time-domain statistics such as mean, standard deviation (SD), maximum, minimum, skewness, kurtosis, total activity time (TAT), and proportional integral mode (PIM). The features extracted from the heart rate consisted of the mean, minimum, maximum, SD of the normal-to-normal (NN) intervals (SDNN), the percentage of successive NN intervals that differ by more than 50 ms (NN50), and the root mean square of successive differences (RMSSD). To identify the most informative features, their importance was assessed using a random forest-based feature importance analysis [21]. We used Scikit-learn's Random Forest classifier to compute feature importance, applying the Gini criterion to quantify each feature's contribution. A forest of 100 trees was trained, and features with importance scores below 0.02 were excluded, as shown in Fig. 1. Based on importance scores, we excluded low-contributing features to improve the performance of the model. For heart rate, the mean, SD, maximum, minimum, SDNN, and RMSSD were included. For actigraphy, ENMO, the mean, SD, maximum, PIM, and TAT were included, and the rest were removed. After feature selection, the feature space was further

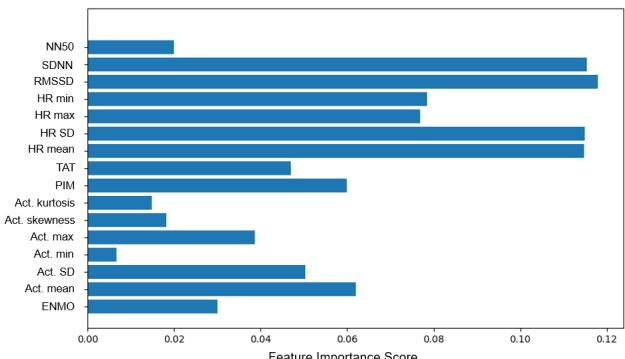

Fig. 1. Feature importance scores for actigraphy and heart rate features

segmented using a 40-epoch window with a stride of one epoch to prepare the input of the model. The reference label of the last epoch in the moving window was used as the target label.

### D. Classification model

As shown in Fig. 2, the proposed model architecture consisted of two main parts: representation learning and temporal learning. Since the actigraphy and heart rate signals have distinct temporal and spectral characteristics, they were fed into separate CNN branches to extract more informative features. Each branch consisted of 3 CNN layers, each CNN layer was followed by a rectified linear unit (ReLU) activation and a dropout. In particular, Conv1D(32,5) refers to using 1D convolution layer with 32 filters and a kernel size of 5. The outputs of the CNN branches were then concatenated and linear fusion was applied using a fully connected layer with ReLU activation, followed by a normalization layer to adaptively weight and integrate the effect of the two modalities. This fused sequence is then fed into a two-layer LSTM encoder, followed by attention pooling. The attention-based layer was included to capture the temporal dependencies and learn the most informative parts of the sequence. In the attention pooling layer, each hidden unit was projected to a scalar attention score, and a softmax was used to normalize the weights. Finally, the context vector, defined as the weighted sum of the hidden states, was computed. To include subject-specific information, subjects' age and sex were concatenated with the context vector and fed to a dense layer as the last layer to generate the assigned classes. This architecture was trained for two-class (wakefulness versus sleep) and three-class (wakefulness, non-REM, and REM) sleep stages.

### E. Ablation Study

We trained seven variants of our selected architecture for the 3-class model with demographics, varying the number of CNN branches (one versus two branches), the fusion method (concatenation, cross-attention, and linear), the input type (engineered features versus raw signals), and the use of attention pooling.

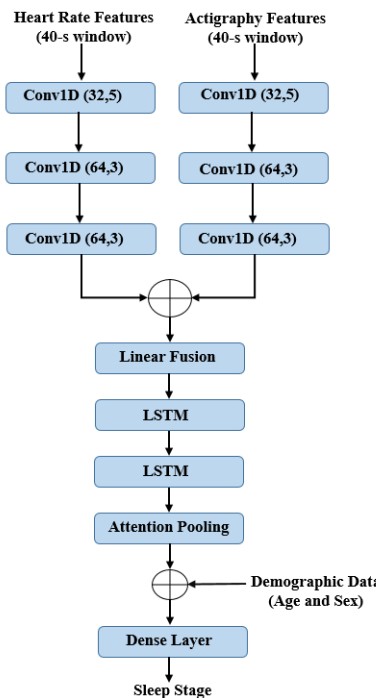

Fig. 2. Overview of the proposed model for two-class and three-class sleep stage classification.

### F. Evaluation and Statistical Analysis

From the detected sleep intervals of the 2-class model, sleep measures including total sleep time (TST), sleep latency (SL), and sleep efficiency (SE) were calculated and compared with the PSG-based measures. For each comparison, Pearson's correlation and p-value were reported. Additionally, we performed subgroup analyses by sex (male vs. female), age ($\leq 65$ vs. $> 65$), and race (White, Black, Hispanic, and Asian), reporting accuracy and weighted $F_1$-score for each subgroup, along with p-value to assess statistical differences across these groups. We also performed a sensitivity analysis across window lengths (10–50 epochs) and compared model performance based on accuracy and weighted $F_1$-score.

## III. RESULTS

Fig. 3 depicts an example trace of the ENMO, heart rate, and the detected sleep stages compared with the PSG-based stages. As can be seen, wakefulness is associated with higher activity counts, quantified with ENMO, and higher heart rate compared to sleep. During REM, heart rate increases relatively compared to non-REM, while activity remains minimal. This section introduces the evaluation metrics and experimental setup, followed by the quantitative results.

### A. Performance Metrics

We evaluate our model's performance using per-class recall across all stages, the overall weighted $F_1$-score, overall accuracy, and Cohen's Kappa coefficient ($\kappa$), which together

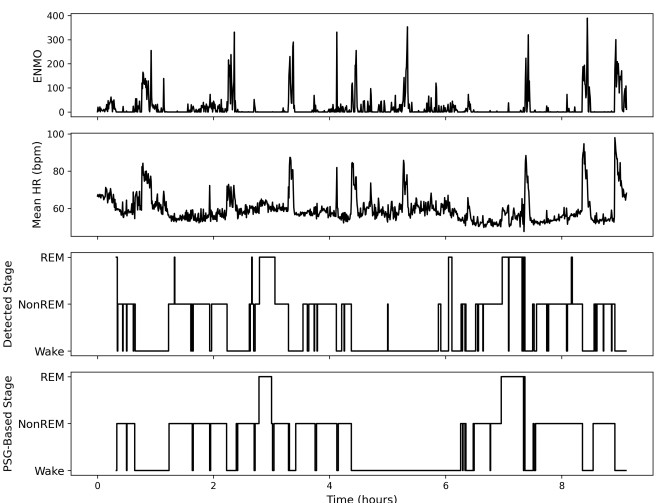

Fig. 3. Example trace of ENMO, heart rate, and the detected sleep stages compared with PSG-based stages for the proposed 3-class sleep stage classification with demographics.

TABLE II
HYPERPARAMETER SEARCH GRID AND CHOSEN VALUES

| Hyperparameter | Range | Selected Value |
|---|---|---|
| Learning rate | $[10^{-4}, 10^{-2}]$ | $1 \times 10^{-3}$ |
| Number of epochs | $[50, 250]$ | 200 |
| LSTM hidden size | $\{128, 256, 512\}$ | 128 |
| Batch size | $\{128, 256, 512\ 1024, 2048\}$ | 512 |
| L2 regularization | $\{0, 10^{-6}, 10^{-5}, 10^{-4}\}$ | 0 |

TABLE III
PERFORMANCE OF OUR BEST MODEL ACROSS DIFFERENT WINDOW SIZES

| Window size | Overall metrics | | 3-class Recall | | |
|---|---|---|---|---|---|
| | Accuracy (%) | Weighted $F_1$ (%) | Wake (%) | Non-REM (%) | REM (%) |
| 10 | 76.50 | 76.75 | 74.48 | 81.72 | 59.63 |
| 20 | 78.58 | 78.78 | 78.74 | 81.85 | 64.51 |
| 30 | 79.12 | 79.38 | 77.33 | **82.91** | 67.46 |
| 40 | **79.60** | **80.00** | **80.10** | 81.60 | **70.00** |
| 50 | 79.52 | 79.80 | 79.39 | 81.95 | 69.68 |

provide a comprehensive assessment of classification performance. The per-class metrics are computed using a one-vs.-rest approach. The accuracy and weighted $F_1$-score are defined as follows:

$$\text{Accuracy} = \frac{1}{N} \sum_{c=1}^{C} TP_c, \tag{1}$$

$$\text{Weighted } F_1 = \frac{1}{N} \sum_{c=1}^{C} n_c \cdot F_1^{(c)}, \tag{2}$$

where $TP_c$ is the number of true positives for class $c$, $N$ is the total number of epochs, $n_c$ is the number of true epochs belonging to class $c$, and $F_1^{(c)}$ is the $F_1$-score of class $c$.

### B. Experimental Setup

Our model was implemented in PyTorch and trained on an NVIDIA Tesla P100-PCIE 12 GB GPU. The extracted data set was randomly divided into training set (75%), validation (15%), and test set (10%). We used the stochastic gradient descent (SGD) optimizer with a learning rate of 0.001 and a batch size of 512, training the model for 200 epochs. The best model parameters were selected based on the validation set. As the classes were highly imbalanced, we used categorical cross-entropy loss with class weights calculated based on the smoothed inverse frequency of each class suggested in [17]. The hyperparameter search grid used to find the optimal values is shown in Table II.

### C. Window-Size Sensitivity Analysis

We performed a sensitivity analysis on window sizes of 10, 20, 30, 40 and 50 epochs for the 3-class model. The results are shown in Table III, confirming that a 40-epoch window achieves the highest performance.

### D. Model Selection via Ablation Study

Table IV shows the overall accuracy, weighted $F_1$, and per-class recall for each of the seven variants. The selected two-branch CNN–LSTM–attention model with linear fusion achieved an overall accuracy of 79.60% using extracted features. For the same model applied to raw inputs, accuracy drops to 78.91%. Using linear fusion increased wake recall by 2.39% and REM recall by 4.07% compared to concatenation fusion. This model was used for all further analyses.

### E. Model Performance and Comparative Analyses

Table V shows the performance of the proposed model for two-class sleep staging in comparison with previous works. Our model with demographics achieved the highest accuracy of 88.19%, weighted $F_1$-score of 88.16%, and Kappa score of 0.72. The model also achieved the highest recall for detecting the wake class (79.77%), while preserving high sleep recall with or without including demographics. As shown in Table VI, our model with demographics achieved the highest overall accuracy (79.60%), Kappa score (0.65), and wake recall (80.10%) for three-class sleep staging compared with previous works. Even without including demographics, our model obtained a higher Kappa score (0.64). These results were achieved while non-REM and REM recalls remained higher or comparable to those of other approaches. The confusion matrices for 2-class and 3-class sleep staging with demographics are shown in Fig. 4. Fig. 5 shows scatter plots of the detected versus PSG-based sleep measures. Strong significant agreements were obtained between the detected and the reference TST (r=0.80, p<0.001) and SE (r=0.76, p<0.001). A moderate significant agreement was between them for SL (r=0.57, p<0.001).

### F. Performance Across Demographic Subgroups

Table VII summarizes the model performance in different demographic subgroups. No significant differences were found across these subgroups.

**TABLE IV**
ABLATION OF INPUT TYPE, FUSION METHOD, AND ARCHITECTURE FOR 3-CLASS SLEEP STAGE CLASSIFICATION

| Variant | Inputs | Fusion | Overall metrics | | 3-class Recall | | |
|---|---|---|---|---|---|---|---|
| | | | Accuracy (%) | Weighted $F_1$ (%) | Wake (%) | Non-REM (%) | REM (%) |
| 1 CNN-branch + LSTM + Attention | Engineered features | Linear | 79.53 | 79.68 | 77.91 | 83.85 | 65.23 |
| 1 CNN-branch + LSTM + Attention | Raw signals | Linear | 78.41 | 78.57 | 75.34 | 83.35 | 64.84 |
| 2 CNN-branches + LSTM + Attention | Raw signals | Linear | 78.91 | 79.01 | 74.69 | **84.42** | 65.72 |
| 2 CNN-branches + LSTM | Engineered features | Linear | 79.45 | 79.72 | 77.46 | 83.19 | 68.43 |
| 2 CNN-branches + LSTM + Attention | Engineered features | Concatenation | 79.56 | 79.71 | 77.71 | 83.85 | 65.93 |
| 2 CNN-branches + LSTM + Attention | Engineered features | Cross attention | 76.73 | 77.08 | 75.89 | 80.67 | 62.26 |
| 2 CNN-branches + LSTM + Attention | Engineered features | Linear | **79.60** | **80.00** | **80.10** | 81.60 | **70.00** |

**TABLE V**
COMPARISON OF OUR MODEL WITH PREVIOUS STUDIES FOR 2-CLASS (WAKE VS. SLEEP) CLASSIFICATION

| Methods | Features | Overall Metrics | | | 2-class Recall | |
|---|---|---|---|---|---|---|
| | | Accuracy (%) | Weighted $F_1$ (%) | $\kappa$ | Wake (%) | Sleep (%) |
| Ensemble [16] | Actigraphy and Heart Rate | 84.40 | 87.60 | 0.64 | 67.00 | **93.00** |
| MLP[a][15] | Actigraphy, Heart Rate, and Clock proxy | 77.40 | — | 0.50 | 72.00 | 80.00 |
| Our work | Actigraphy and Heart Rate | 87.85 | 87.83 | 0.71 | 79.63 | 91.52 |
| Our work | Actigraphy, Heart Rate and Demographics | **88.19** | **88.16** | **0.72** | **79.77** | 91.95 |

[a] MLP: Multi Layer Perceptron

**TABLE VI**
COMPARISON OF OUR MODEL WITH PREVIOUS STUDIES FOR 3-CLASS SLEEP STAGE CLASSIFICATION

| Methods | Features | Overall Metrics | | | 3-class Recall | | |
|---|---|---|---|---|---|---|---|
| | | Accuracy (%) | Weighted $F_1$ (%) | $\kappa$ | Wake (%) | Non-REM (%) | REM (%) |
| Seq2Seq LSTM[a][17] | Actigraphy and Heart Rate | 79.11 | **80.00** | — | 78.00 | 81.80 | 70.90 |
| Ensemble [16] | Actigraphy and Heart Rate | 78.20 | 69.80 | 0.62 | 75.00 | **84.00** | 42.00 |
| MLP[b][15] | Actigraphy, Heart Rate and Clock proxy | 72.30 | — | 0.28 | 60.00 | 65.10 | 65.00 |
| Our work | Actigraphy and Heart Rate | 78.81 | 79.19 | 0.64 | 79.74 | 80.35 | **70.25** |
| Our work | Actigraphy, Heart Rate and Demographics | **79.60** | **80.00** | **0.65** | **80.10** | 81.60 | 70.00 |

[a] Sequence-to-Sequence LSTM
[b] MLP: Multi Layer Perceptron

## G. Model Interpretability

Fig. 6 visualizes the 128-dimensional context vectors in two dimensions using t-distributed stochastic neighbor embedding (t-SNE). By projecting each vector into a 2D space, we observe distinct clusters for Wake, non-REM, and REM stages, indicating that the model effectively learned to capture meaningful differences between sleep stages.

## IV. DISCUSSION

In this study, we used 1012 concurrent wrist actigraphy and PSG recordings, from which heart rate and sleep staging scores were derived, to develop and validate a subject-specific, attention-based learning model for both 2-class (wakefulness versus sleep) and 3-class (wakefulness, non-REM and REM) sleep staging. Key attributes of the model include: 1) improved overall accuracy, Kappa score and wake recall compared to many other previously proposed models; 2) integration of demographics as subject-specific measures; and 3) accurate representation of sleep measures compared to PSG-based sleep measures. The sleep data are highly imbalanced, with more sleep intervals than wakefulness. Between sleep stages, non-REM intervals outnumber REM intervals. Therefore, a

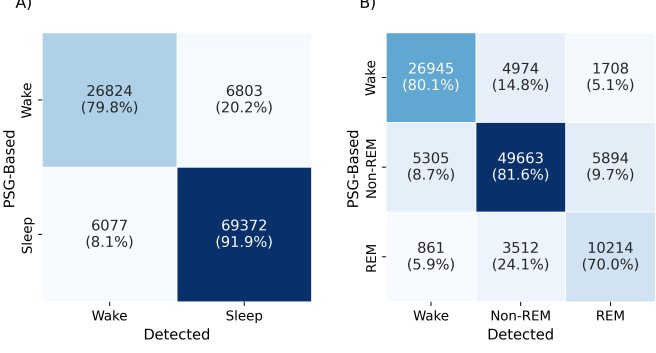

Fig. 4. Confusion matrices for sleep staging when including demographic information in the model: A) 2-class (Wake vs. Sleep) and B) 3-class (Wake, Non-REM, REM).

challenge in sleep staging is to achieve high recall for wakefulness and REM. To address class imbalance in this work, we used categorical cross-entropy loss with class weights calculated based on the number of intervals in each class [17]. Experimental results showed that our model achieved the highest accuracy among different approaches, with significantly

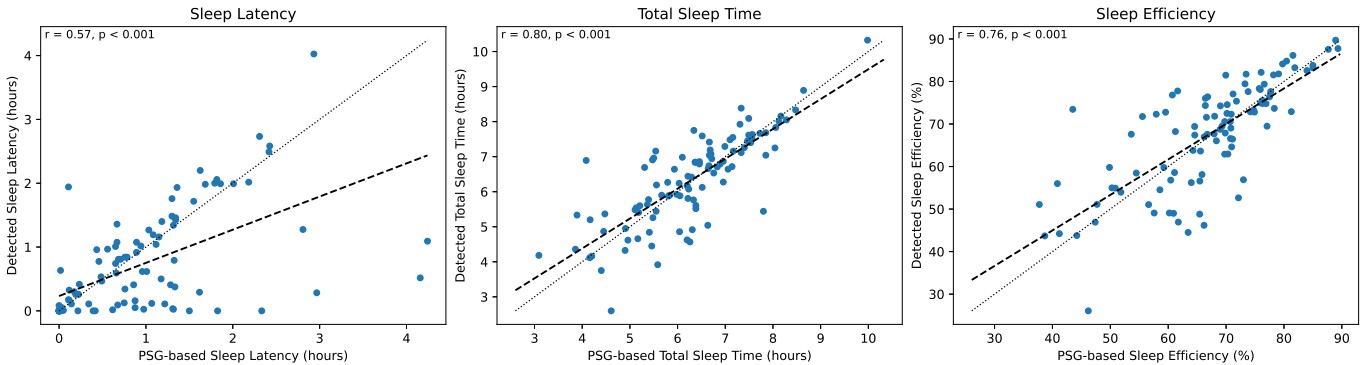

Fig. 5. Scatter plots of predicted vs. reference values for sleep latency, total sleep time, and sleep efficiency. The dotted line represents the unity line, and the dashed line represents the regression line.

TABLE VII
PERFORMANCE BY DEMOGRAPHIC SUBGROUPS

| Metric | Sex | | | Age | | | Ethnicity | | | | |
|---|---|---|---|---|---|---|---|---|---|---|---|
| | Male | Female | p-value | 65 or Below 65 | Above 65 | p-value | White | Black | Hispanic | Asian | p-value |
| Accuracy (%) | 78.40 | 80.94 | NS[a] | 80.34 | 79.05 | NS | 78.87 | 80.21 | 82.15 | 73.60 | NS |
| Weighted $F_1$ (%) | 78.37 | 81.11 | NS | 80.50 | 79.04 | NS | 78.74 | 80.23 | 82.52 | 74.13 | NS |

[a] NS: not significant.

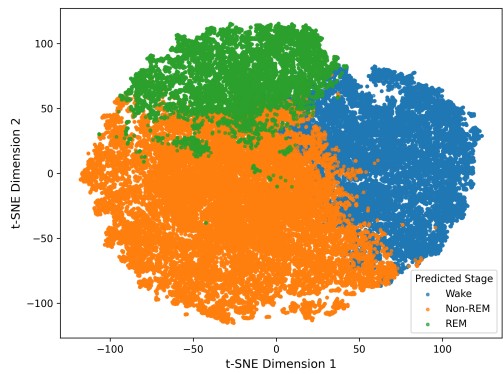

Fig. 6. T-distributed Stochastic Neighbor Embedding (t-SNE) visualization of context vectors.

improved wake sensitivity while non-REM and REM recall remained comparable to those of other methods.

Previous works, such as [17], fed their models with raw actigraphy and heart rate signals and let CNN layers extract informative latent representations. In this work, we implemented several strategies: 1) hand-engineered features were extracted for both actigraphy and heart rate, and 2) the importance of the features were assessed by random forest and features with lower importance scores were excluded; 3) separate CNN branches were designed to analyze the features of actigraphy and heart rate; and 4) linear fusion was applied to the latent space of the two modalities. Our results indicates that these strategy boosted Kappa score by at least 0.1 in two-class and 0.02 in three-class tasks.

The characteristics of the brain activity during sleep differ between sexes and age groups [22]. Therefore, the subjects' age and sex information was extracted from the provided medical records and included in the model. Such subject-specific data were directly concatenated with the attention outputs and were fed into a dense layer to generate the detected classes. We observed that including age and sex in the model boosted its performance in terms of all metrics for the 2-class task and for almost all metrics in the 3-class task (except REM recall).

The model was trained for both 2-class and 3-class sleep staging. With the 2-class output, we could provide an estimate of sleep duration, quantified by TST and SE, whereas the 3-class sleep staging provides detailed insight into sleep depth. Detecting the intervals of REM sleep is neurologically informative. For instance, experiencing delayed onset of REM sleep is associated with Alzheimer's disease [23]. Detecting REM sleep behavior disorder, in which individuals move their bodies in response to dreams, could assist in the diagnosis of Lewy body dementia [24] or Parkinson's disease [25].

One limitation of this study is that the model was developed and evaluated on one database, and its generalizability on other databases is yet to be evaluated in our future work. Despite the high Cohen's kappa that this model achieves (0.72), our model can still mistakenly predict transitions from wake to REM in 0.5% of epochs. In the future, we will explore graphical models to enforce realistic sleep stage transitions, making the predictions more physiologically plausible.

## V. CONCLUSION

This study proposed an attention-based model that achieves higher overall accuracy and Kappa score to detect intervals

of wakefulness versus sleep stages using actigraphy and heart rate. The performance of the model was further enhanced by extracting hand-engineered features and combining subject-specific demographics with the attention output. This work leverages wrist-worn wearables, specifically with embedded accelerometry and PPG, by proposing a subject-specific model that accurately detects wakefulness and estimates the duration and depth of sleep. Such robust wearable technology offers a more accessible home-based sleep monitoring device, which can be used in populations such as children or older adults who cannot tolerate in-laboratory sleep tests or those living in remote areas.

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
