# OpenReview forum: "Sleep Stage Detection from Actigraphy and Heart Rate Using an Attention-Based Model"
_IEEE.org/EMBS/BHI/2025/Conference — BHI 2025_

### Official Review · Reviewer_QChZ · 2025-07-03
**Sleep Stage Detection from Actigraphy and Heart Rate Using an Attention-Based Model**

**Confidence:** 5
**Clarity Of Writing:** good
**Clinical Significance:** good
**Methodological Novelty:** good
**Overall Rating:** 7

**Experiments And Results:**

good

**Questions For The Authors:**

1.Rationale for Model Architecture Choices: The paper uses a two-branch CNN followed by LSTM with attention. Could you elaborate on why this specific architecture was chosen? What are the key advantages of using separate CNN branches for actigraphy and heart rate, and how does the LSTM with attention build upon these representations to improve sleep stage classification? What happens and what are your hypothesis for an approach that has been tested recently?

2.Detailed Explanation of Feature Importance Analysis:Could you provide more detail on the random forest-based feature importance analysis? What specific criteria were used to determine the "importance" of each feature, and what was the threshold used for excluding low-contributing features? It is important that your methods is clear.

**Strengths:**

-Clear Objective: The paper clearly defines its objectives: improving REM detection accuracy and designing a model to assess the contribution of subject-specific demographics.
Well-Justified Approach: The paper effectively justifies the use of actigraphy and heart rate as more accessible and scalable alternatives to PSG.
-Dataset and Preprocessing:
The use of the MESA dataset, a well-established and diverse dataset, is a strength.
The data preprocessing steps are clearly described and seem appropriate.
-Feature Extraction and Selection:
The choice of features (ENMO, TAT, PIM, SDNN, RMSSD, etc.) is well-motivated.
The feature selection process using random forest-based feature importance analysis is a sound approach.
-Model Architecture:
The two-branch CNN architecture for actigraphy and heart rate is a reasonable design choice.
The use of LSTM with attention for temporal learning is well-justified.
-Evaluation and Comparison:
The paper uses appropriate evaluation metrics (accuracy, weighted F1-score, Kappa) for a multi-class classification problem.
The model is compared with previous studies, providing context for its performance.

**Summary Of The Paper:**

This study proposes deep learning models for detecting wakefulness versus different stages of sleep using heart rate and wrist actigraphy data from the MESA sleep dataset. Two sets of features are extracted from heart rate and actigraphy and separately fed into two CNN branches, then merged and fed to a deep classifier. The best model achieves 88.19% accuracy for wakefulness vs. sleep and 79.6% accuracy for wakefulness, non-REM, and REM sleep stage classification. The integration of heart rate, actigraphy signals, and demographic data in a deep framework improves sleep stage staging performance. This work offers a subject-specific approach for sleep assessment based on convenient wearables.

**Weaknesses:**

-Modelling:

You are taking the data as a two branch CNN and then merging it into a 1D with dense Layer, wouldn't it be more effective to keep it as a two branch model?
-Missing Rationale for specific Model and Justification:

You should have provided a better justification for choosing that modelling specifically. And add why are expecting such results.

---

### Official Review · Reviewer_PG1g · 2025-07-15
**Sleep Stage Detection from Actigraphy and Heart Rate Using an Attention-Based Model**

**Confidence:** 4
**Clarity Of Writing:** good
**Clinical Significance:** excellent
**Methodological Novelty:** great
**Overall Rating:** 7

**Experiments And Results:**

good

**Questions For The Authors:**

1. Is MESA the only dataset available for sleep data?
2. Are the other methods in literature mentioned in your paper also trained and tested on MESA database?
3. Do you think the model is generalizable to data collected from patients in another dataset or collected from another device?

**Strengths:**

- The premise is sound
-  The performance of the models is very good. I think the analysis of the machine learning model you provide is convincing.
-  I like the emphasis on feature importance. I think it provides a useful insight into useful metrics for sleep performance.
- I appreciate the discusssion section provided which goes into how sleep tracking could be instrumental in tracking neurodegenerative diseases.

**Summary Of The Paper:**

This paper presents a deep learning model that accurately tracks sleep stages using wearable data—specifically heart rate and wrist movement. The authors developed a new architecture that’s particularly good at detecting REM sleep, a common weak spot for other methods. By combining sensor data with demographic information, their model offers a more precise and accessible way to monitor sleep at home, overcoming the need for clinical polysomnography.

**Weaknesses:**

-  In Table 3, please provide more details on what methods [15] and [16] are. Mention briefly what architecture they are using? Are [15] and [16] the gold standard in wearable sleep stage detection?
- Commentary on how [15] and [16] lack would build a more convincing argument. This commentary is crucial to convincing the reader of your work.
- Section 2:D. Are you talking about R peaks in the QRS complex?

---

### Official Review · Reviewer_5tLS · 2025-07-17
**Sleep Stage Detection from Actigraphy and Heart Rate Using an Attention-Based Model**

**Confidence:** 4
**Clarity Of Writing:** good
**Clinical Significance:** good
**Methodological Novelty:** fair
**Overall Rating:** 4
**Final Rating:** 5

**Experiments And Results:**

fair

**Questions For The Authors:**

- The proposed model architecture is primarily a simple combination of standard components, lacking significant structural novelty. Could the authors explicitly explain why their proposed fusion strategy is theoretically or practically superior to established methods such as cross-attention or multimodal transformers?

- The authors adopt a sliding window with stride = 1, where each 40-epoch input corresponds to a single label, resulting in a large number of highly overlapping samples. This design is prone to information leakage between training and validation sets or redundant gradient updates during training. Can the authors provide a sensitivity analysis of model performance with respect to the stride parameter?

- The demographic vector is simply concatenated after the attention module. This approach fails to effectively model the dynamic modulation effect of demographic features on the temporal sequence and lacks deeper modeling logic. Could the authors explore more expressive fusion methods to better capture the influence of demographic factors on sleep stage classification?

- The current model processes data sequentially without considering the influence of past sleep stages on the current state. For example, it is physiologically unlikely to transition directly from wakefulness to REM. Could the authors consider incorporating structures like a Conditional Random Field (CRF) layer to model label transition dependencies and enhance physiological plausibility and prediction consistency?

- Could the authors include attention weight visualizations to illustrate which signal regions or features contribute most to the classification of different sleep stages? This would enhance both the interpretability and clinical credibility of the model.

**Strengths:**

- The model is trained on the MESA dataset with 1,012 subjects, providing strong practical relevance and applicability.

- The two-branch CNN architecture is well designed, the attention mechanism enhances interpretability, and the innovative integration of demographic information is empirically shown to improve model performance.

- The study conducts a comprehensive evaluation on both 2-class and 3-class sleep stage classification tasks, using a wide range of evaluation metrics.

**Summary Of The Paper:**

The authors utilize heart rate signals and actigraphy data as inputs to a two-branch CNN to extract the main features of each modality. The fused features are then passed into LSTMs to capture temporal features. After the attention pooling extracts the key temporal features, they are concatenated with demographic data and finally used for sleep stage classification. The model demonstrates better performance compared to baselines, with higher overall accuracy, Kappa score and wake recall.

**Weaknesses:**

The paper currently lack of structural novelty since it is based on simple fusion of demographic information; additionally there is limited consideration done towards the interpretability aspect of their pipeline. More details can be found in the questions section.

---

### Official Review · Reviewer_dRZ9 · 2025-07-21
**a novel attention-based deep learning model for sleep stage classification using wearable-derived heart rate and wrist actigraphy data.**

**Confidence:** 4
**Clarity Of Writing:** good
**Clinical Significance:** great
**Methodological Novelty:** good
**Overall Rating:** 4
**Final Rating:** 5

**Experiments And Results:**

good

**Questions For The Authors:**

- Could you clarify the choice of hand-engineered features over raw data end-to-end learning?
- How does the model perform across different demographic subgroups (age, sex, ethnicity)?
- Have you considered or tested your model on other wearable datasets or with different sensors?
- The motivation of the current paper closely resembles some prior work such as AccSleepNet, which also proposed a deep learning architecture for sleep stage classification using wrist-worn accelerometer data, aiming to provide a scalable alternative to PSG.

**Strengths:**

- Use of Easily Accessible Data Sources: The reliance on heart rate and wrist actigraphy data, rather than full polysomnography (PSG), is a major practical advantage. These signals are commonly collected by commercial wearable devices, making the proposed method highly scalable and non-invasive for large-scale, at-home sleep monitoring. This significantly improves the feasibility of real-world deployment, especially in resource-limited or population-scale settings.
- Subject-Specific Modeling: Incorporating demographic variables (age, sex) into the model pipeline is a valuable step to personalize predictions and enhance performance.

**Summary Of The Paper:**

The manuscript proposes a novel attention-based deep learning model for sleep stage classification using wearable-derived heart rate and wrist actigraphy data. Leveraging data from the large, multi-ethnic MESA cohort, the authors extract hand-engineered features and design a dual-branch CNN to separately process heart rate and actigraphy inputs, which are then fused and fed into an LSTM with attention. The model integrates subject demographics to improve classification. Results show promising accuracy and Kappa improvements in both two-class (wake vs sleep) and three-class (wake, non-REM, REM) sleep staging, compared to prior works.

**Weaknesses:**

- Limited Novelty in Model Architecture: The model components (CNN, LSTM, attention) are standard deep learning building blocks in time-series biosignal analysis. The novelty mainly resides in the specific application and integration strategy rather than fundamentally new methodology.
- Feature Engineering vs. Raw Data Learning: The rationale for using hand-engineered features instead of end-to-end raw signal learning could be elaborated further. Recent deep learning approaches often outperform engineered features by learning richer latent representations directly. A comparison or ablation between these approaches is lacking.
- Limited Exploration of Demographics Impact: While demographics are integrated, the manuscript does not deeply analyze how different demographic groups affect performance or whether the model exhibits bias across subgroups.
- Lack of Full 5-Stage Sleep Classification: The model is evaluated only on binary (wake vs. sleep) and three-class (wake, REM, NREM) classification tasks. While this simplification is common in wearable-based studies, it limits clinical utility. The absence of fine-grained 5-stage sleep classification (W, N1, N2, N3, REM)—which is the standard in PSG-based scoring—reduces the potential for the model to fully support clinical diagnostics or detailed sleep architecture analysis.